# An Atypical Course of Cardiomyopathy Syndrome (CMS) in Farmed Atlantic Salmon (*Salmo salar*) Fed a Clinical Nutrition Diet

**DOI:** 10.3390/microorganisms12010026

**Published:** 2023-12-22

**Authors:** Johan Rennemo, Kjetil Berge, Muhammad Naveed Yousaf, Tommy Berger Eriksen, Eirik Welde, Camilla Robertsen, Bjarne Johansen, Charles McGurk, Espen Rimstad, Erling Olaf Koppang, Håvard Bjørgen

**Affiliations:** 1Skretting AS, 4016 Stavanger, Norway; johan.rennemo@skretting.com (J.R.); kjetil.berge@skretting.com (K.B.); 2Skretting Aquaculture Innovation (AI), 4016 Stavanger, Norway; muhammad.naveed@skretting.com (M.N.Y.); tommy.berger.eriksen@skretting.com (T.B.E.); charles.mcgurk@skretting.com (C.M.); 3Nordlaks Havbruk AS, 8455 Stokmarknes, Norway; eirik.welde@nordlaks.no (E.W.); camilla.robertsen@nordlaks.no (C.R.); bjarne.johansen@nordlaks.no (B.J.); 4Unit of Virology, Faculty of Veterinary Medicine, Norwegian University of Life Sciences, 1432 Ås, Norway; espen.rimstad@nmbu.no; 5Unit of Anatomy, Faculty of Veterinary Medicine, Norwegian University of Life Sciences, 1432 Ås, Norway; erling.o.koppang@nmbu.no

**Keywords:** clinical nutrition, CMS, fish health, fish welfare, PMCV, population medicine

## Abstract

Cardiomyopathy syndrome (CMS) poses a significant threat to farmed Atlantic salmon (*Salmo salar*), leading to high mortality rates during the seawater phase. Given that controlled experimental challenge trials with PMCV do not reproduce the mortality observed in severe field outbreaks of CMS, field trials on natural CMS outbreaks are warranted. This field study explored the impact of a clinical nutrition intervention, specifically a diet enriched with eicosapentaenoic acid (EPA) and docosahexaenoic acid (DHA), on a severe CMS outbreak in a commercial sea farm. CMS was diagnosed in a single sea cage with high mortality rates. Histopathological analysis, RT-qPCR in situ hybridization for virus detection, and fatty acid composition analysis were used to monitor the impact of disease and the inclusion of EPA and DHA in heart tissue. Following the implementation of clinical nutrition, a decline in mortality rates, regression of CMS-associated changes, and a significant reduction in piscine myocarditis virus (PMCV) RNA load were observed within the salmon population. Fatty acid composition analysis of heart samples demonstrated increased levels of EPA and DHA, reinforcing the association between dietary factors, viral load dynamics, and overall fish health. Although further validation is needed in future studies, as field trials may not be sufficient to establish causation, our results indicate that optimizing the EPA + DHA levels may prove beneficial in severe CMS outbreaks.

## 1. Introduction

Cardiomyopathy syndrome (CMS) is a viral disease in farmed Atlantic salmon (*Salmo salar*) and is associated with infection with piscine myocarditis virus (PMCV), a non-enveloped virus with a double-stranded RNA genome with the characteristics typical for members of the family *Totiviridae* [1,2]. Initially identified in farmed Atlantic salmon in Norway [3], CMS has also been documented in Scotland [4], the Faroe Islands [5], Ireland [6], and Canada [7]. CMS is considered the most common cause of mortality during the seawater phase [8], has a significant impact on fish health and welfare and the economy of the industry, and is thus a major concern in Atlantic salmon aquaculture [9].

Typically, CMS outbreaks commonly occur following stressful events such as handling, net-cleaning, lice treatment, weather changes, or other diseases. The disease predominantly affects large salmon in their second year at sea, often resulting in a sudden increase in the mortality of fish that were previously clinically healthy [9]. Affected fish exhibit circulatory failure, occasionally accompanied by external signs such as exophthalmia, ventral skin haemorrhages, and oedematous skin scale pockets [9]. Autopsy examinations reveal ascites, liver changes, atrial rupture, or blood-filled pericardial cavities [9]. The presence of characteristic histopathological changes, in particular subendocardial myocarditis in the atrial and ventricular spongiosum, are used to confirm the diagnosis. Supportive laboratory findings are reverse transcription-quantitative polymerase chain reaction (RT-qPCR) for PMCV. While controlled challenge experiments have demonstrated virus transmission after i.m. injection of material containing virus [2,10], the localized nature of CMS outbreaks in individual cages in a production setting raises questions regarding the contagiousness of the disease [11]. There are limited interventions available when CMS is confirmed. The mortality rates usually remain elevated in affected cages, which necessitates early slaughter to maintain animal welfare standards and minimize economic losses.

The limited clinical tools for managing CMS, as well as for other viral diseases occurring in the seawater phase of Atlantic salmon farming, have brought attention to alternative prophylactic health strategies [12]. Currently, there is no available vaccine against CMS, but smolt that is genetically selected for CMS resistance (CMS-QTL) is commonly used. Fish farmers often use functional feeds during high-risk periods for CMS, leveraging general immune modulation to mitigate the disease’s impact. However, the mechanisms underlying the functionality of such diets are often poorly understood. Martinez-Rubio et al. [13] investigated the effects of a clinical nutrition diet with low lipid content but relatively high levels of eicosapentaenoic acid (20:5n-3; EPA) in small fish (±150 g) challenged with PMCV. Their study revealed that the altered lipid composition was associated with significantly reduced histopathological changes in the heart and liver, as well as decreased viral RNA loads at 8 weeks post-challenge. Moreover, the genes related to T cell immunity exhibited lower expression levels compared to a control group fed a standard diet. Although this trial employed an extreme diet with unusual composition, the findings highlight the potential of clinical nutrition in the mitigation of the effects of PMCV infection in farmed salmon.

There is a large diversity among individual fish in their response to PMCV infection, where some fish eliminate the virus infection while others experience significant viral replication and extensive damage to heart tissue [10]. The variability in resistance to PMCV is believed to be genetically influenced, and investigations have revealed a strong genetic component for resistance against PMCV and identified significant quantitative trait loci (QTLs) [14]. In a recent study by Mogahadam et al. [15], a QTL with strong resistance to PMCV was described on chromosome 23, in proximity to delta-5 fatty acyl desaturase and fatty acid desaturase 2 genes, both playing a role in the production of polyunsaturated fatty acids (PUFA). This might indicate that genetic control of PUFA production is of importance for PMCV resistance, and a link between PUFA and resistance to infectious diseases, generally, has been suggested earlier [16]. A natural follow-up to this would be to study if dietary changes in PUFA quality and levels should be equally important in PMCV resistance.

In this descriptive field study, we report findings from the use of clinical nutrition in large Atlantic salmon (approximately 2.5 kg and fed until slaughter weight) affected by a CMS outbreak at a typical commercial sea farm. Using RT-qPCR and in situ hybridization for PMCV detection, histopathological analysis, and lipid composition analysis of the heart, we demonstrate a decline in mortality, overall regression of CMS-related changes, and reduction of viral RNA load within the population. We found a significantly altered fatty acid composition in the heart in fish fed the clinical nutrition diet. The findings are significant as they contrast the typical progression of severe CMS outbreaks, where premature slaughter is the only economically viable option. Although further validation is needed in future studies, as field trials may not be sufficient to establish causation, our results emphasize the importance of optimizing the EPA + DHA levels to mitigate CMS and thus generally improve fish health in the later stages of the marine growth phase. 

## 2. Materials and Methods

### 2.1. Fish Stock and Sampling Procedure

In August 2020, a total of approximately 800,000 smolts of the Aquagen strain, with an average weight of 238 g, were transferred to sea cages located in Hadselfjorden in Northern Norway owned by Nordlaks Oppdrett AS. Prior to sea transfer, all fish were vaccinated with Pentium Forte Plus Vet. (Elanco Denmark APS, Bergen, Norway Branch) and Aquavac 6 Vet (MSD Animal Health Norge AS, Bergen, Norway). CMS was first suspected in Cage 3 (C3) in December 2020. CMS-related mortality started to increase from week 14 in 2021, prompting a change in the diet for the fish in C3 from week 23, which was continued until slaughter (see “Nutrition” for more details).

Throughout the production period, C3 was closely monitored, and three major samplings were conducted: the first sampling was performed after CMS had been diagnosed but the cage population still experienced increasing mortality (7 June 2021, week 23), the second sampling took place four weeks later (5 July 2021, week 27), and the third sampling was conducted at the time of slaughter (30 October 2021, week 44). Each sample included ten fish. All sampled fish were euthanized using a lethal dose of anaesthetic (Benzoak vet., ACD Pharmaceuticals AS, Leknes, Norway). Following euthanasia, the fish were autopsied, and samples from the ventricle were collected in RNAlater for further analysis. Heart samples, including the atrium, ventricle, and bulbus arteriosus, were also collected and preserved in 10% neutral buffered formalin for histological analysis. Additionally, a pooled sample of hearts from ten fish in each cage was frozen and stored for the analysis of fatty acid composition. For comparison, fish in cage 2 (C2), unaffected by CMS, were included in the first and second samplings, with a sampling of heart tissue for RTq-PCR and fatty acid composition analysis. Each sampling included ten fish. The samplings were performed in accordance with national regulations for animal welfare (Forskrift om drift av akvakulturanlegg §34. Avlivning av fisk).

### 2.2. Analysis of Production Data

Production data, i.e., appetite- and mortality numbers, were monitored daily in both C2 and C3. Appetite was registered, while mortality numbers were accounted for by daily registrations from fish farm personnel. Mortality was reported as a weekly percentage of the total number of fish in the respective cages. Seawater temperatures at 5 m below sea level were 7.8 °C, 10.0 °C, and 8.0 °C in weeks 23, 27, and 43, respectively.

### 2.3. RT-qPCR—Piscine Myocarditis Virus (PMCV)

PMCV RNA was measured by RT-qPCR (performed by PatoGen AS, Ålesund, Norway) in all heart samples from both C2 and C3. The RT-qPCR analysis is accredited and validated to ISO17025 standards. Details of the purification method and PCR conditions are not disclosed by PatoGen due to issues related to competing patents. Samples were defined as positive when having a PMCV Ct lower than 37.0.

### 2.4. Histological Analysis

All the samples collected in formalin from C3 were embedded in paraffin wax and stained with haematoxylin and eosin according to standard protocol [17]. Scoring of the heart lesions by histopathological examination was performed as described earlier [10]. In brief, scores 0 and 1 were considered normal, without any histopathological findings (score 0), or single or few focal lesions (score 1). Score 2 had several distinct lesions and increased leukocyte infiltration. Score 3 represented multifocal to confluent lesions in > 50% of tissue and moderate to severe leukocyte infiltration. The atrium and the ventricle were scored separately.

### 2.5. In Situ Hybridization (ISH)—Piscine Myocarditis Virus (PMCV)

All the paraffin-embedded samples from C3 were analyzed for the presence of PMCV RNA using RNAscope^®^ 2.5 HD Singleplex Red Chromogenic Reagent kit (Advanced Cell Diagnostics Inc., Newark, CA, USA) according to the manufacturer’s protocol. The PMCV probe A set of 20 pairs of PMCV probes ‘V-piscine-myocarditis-ORF2’ (cat.no. 555231; Advanced Cell Diagnostics Inc.), targeting the PMCV capsid gene area at nt 1050–1757 bp on GenBank reference JQ728724.1, were designed by the manufacturer using custom software as described by [18]. Positive and negative controls were performed with probes targeting salmon housekeeping gene PPIB and DAPb targeting irrelevant bacterial RNA, respectively. Also, infected and uninfected salmon from a challenge trial with PMCV were used to validate the PMCV probe (see Appendix A for control images).

### 2.6. Image Analysis of In Situ Hybridization

The ISH slides were digitized using a slide scanner (Panoramic Scan II, 3DHistech, Budapest, Hungary) with a 20× magnification objective. An image analysis software equipped with artificial intelligence (Visiopharm version 2022.11 equipped with AI Author architect module) was employed to obtain quantitative data on the ISH labelling of PMVC. The neural network (DeepLabv3+) was trained on user-defined examples of positive ISH labelling to produce an algorithm for automated image analysis, enabling the calculation of positive signal area, i.e., PMCV RNA, and remaining tissue area. The data output from the automated image analysis could be visualized and validated to accurately classify the ISH labelling of PMCV (Appendix A).

The virus area fraction was calculated and expresses the tissue area made up by positive ISH labelling:$$\mathrm{Virus}\;\mathrm{area}\;\mathrm{fraction}\;\left(\%\right)=\frac{\mathrm{Virus}\;\mathrm{area}\;}{\mathrm{total}\;\mathrm{tissue}\;\mathrm{area}}*100$$

The ventricle and atrium of the hearts were manually annotated as different compartments to allow for separate calculations (Appendix A). Note that the lumen in both compartments and the stratum compactum in the ventricle are included in the total tissue area, resulting in a low virus area fraction. The results only provide the ratio between PMCV RNA and total tissue, i.e., not specific levels of virus RNA in the cardiac tissue.

### 2.7. Fatty Acid Composition in Heart Tissue

The fatty acid composition of hearts in both C2 and C3 was determined after the separation of the methyl esters in a gas chromatograph (Scion 436 GC with CP-8400 autosampler, Scion Instruments, Livingstone, UK), equipped with PTV split/splitless injector (70 °C for 2 min, 30 °C/min to 150 °C, 4.0 °C/min to 225 °C and held for 4.58 min), a CP Wax 52 CB capillary column (L: 25 m, id: 0.25 mm, OD: 0.36 mm, df: 0.20 µm), a flame ionization detector, and hydrogen as carrier gas. The fatty acids were identified by retention time using standard mixtures of methyl-esters (Nu-Chek, Elyian, MN, USA), and the fatty acid composition (area %) was determined. All samples were integrated using the software Chromeleon^®^ version 7.2 connected to the GC.

### 2.8. Statistical Analysis

RT-qPCR of PMCV and the histological scoring were analyzed. The differences in histological scorings of the atrium and ventricle between the different time points were tested by a Chi-square test. Means of PMCV virus at the different time points were compared using the Student’s *t*-test. A connecting letters report indicates differences between the time points. Levels not connected by the same letter are significantly different.

### 2.9. Nutrition

Both the C2 and C3 populations were initially fed a standard grower diet in the seawater phase. The diet was changed to *Aqura* (Skretting AS, Stavanger, Norway) in C3 in week 23, 2021. *Aqura* was fed throughout the rest of the production period in C3. *Aqura* differs from standard grower diets, having lower lipid content, higher protein content, and higher marine content. The minimum level of EPA + DHA is 14% of total fat in the feed compared to the standard grower diet which contains typically ~6% (utilized for the fish size relevant for this study). 

## 3. Results

### 3.1. Analysis of Production Data (Cage 2 and 3)

Mortality rates were measured as a percentage of the total number of fish in each cage per week (Figure 1). Cage 2 showed negligible mortality throughout the entire observation period. In C3, mortality slightly increased following the sea transfer and steadily rose to approximately 1.5% weekly mortality by week 24. Feeding with a clinical nutritional diet (*Aqura*) in Cage 3 started in week 23, and from week 24 and onwards, mortality decreased, except for a minor peak observed during weeks 32–37, which coincided with the fish being transferred to another location. Subsequently, after week 37, mortality continued to decline and remained negligible during the final 7 weeks leading up to slaughter. Appetite was unaffected in both cages during the entire observation period.

### 3.2. RT-qPCR—PMCV (Cage 2 and 3)

At the first sampling, only two of ten fish in C2 were positive for PMCV by RT-qPCR and then only with high Ct values (Figure 2). At the second sampling, all fish in C2 tested negative for PMCV. The fish in C2 remained healthy and did not experience any increased mortality until they were slaughtered.

In C3, at the first sampling, the median Ct value of RT-qPCR for PMCV was 26.75 and 24.54 at the second sampling. This indicated a relatively high viral load. At the third sampling, five out of ten fish tested negative for PMCV, and the median Ct value was 34, suggesting a significantly lower viral load compared to the two earlier samplings.

### 3.3. Histopathological Scoring—Hearts from Cage 3

The hearts of fish in C3 were assessed using a scoring system ranging from 0 to 3, with separate scores assigned to the atrium and ventricle (Figure 3). At the first and second samplings, the average scores were similar in both the atrium and ventricle. Overall, the ventricle scored slightly higher. However, there was considerable individual variation among the fish. In the third sampling, there was a clear tendency towards decreased score in the atrium, from 1.7 in to 0.8 (not statistically significant, *p* = 0.07). A decline was also observed in the ventricle from 2.1 to 1.4 (not statistically significant, *p* = 0.39).

### 3.4. In Situ Hybridization (ISH)—Piscine Myocarditis Virus (PMCV) in Cage 3

In C3, ISH was performed on hearts from all fish across the three samplings. Figure 4 illustrates the typical staining patterns corresponding to different histopathological scores. These staining patterns align with the findings reported by Fritsvold et al. [11]. Controls for the ISH runs are shown in Appendix A.

To quantify the presence of PMCV RNA in the heart sections, the area fraction virus (%) was calculated for each section (Figure 5). The results are consistent with the RT-qPCR data, showing the highest average observed at the second sampling across all cardiac compartments. At the third sampling, the presence of PMCV RNA was present but in low amounts in both the atrium and ventricle.

### 3.5. Fatty Acid Profile in Heart Tissue

EPA, DHA, total n-3 and total n-6 levels were measured in heart tissue in both cages (Figure 6). At the first sampling, before clinical nutritional diet (*Aqura*) feeding started, the levels of EPA, DHA, sum n-6 and n-6/n-3 ratio were similar in both pens. After four weeks of clinical nutritional diet feeding in C3, at the second sampling, there was an elevated level of both EPA and DHA in heart tissue and a marked decrease in total n-6 and a correspondingly lower n-6/n-3 ratio. After approximately twenty weeks (third sampling) of clinical nutritional feeding, these changes were even more pronounced.

## 4. Discussion

In this study, we investigated the progression of a typical, severe outbreak of CMS in a population of farmed Atlantic salmon in a single sea cage. Throughout the period from the onset of the CMS outbreak until slaughter, we closely monitored and analyzed the fish. Severe outbreaks of CMS often result in accumulated high mortality rates, and CMS is a significant threat to fish health and welfare as well as causing economic pain. In an attempt to address this issue, we implemented a radical and long-lasting change in diet, i.e., the transition from a standard grower diet to a clinical nutrition diet enriched with elevated levels of EPA and DHA comprising more than 14% of the total fat content. Fatty acid profiling on the hearts demonstrated a significant impact resulting from the dietary change. Subsequently, this was followed by a decline in mortality, a decrease in the histopathological score of the hearts, and a reduction in the fish population’s load of PMCV RNA. The results indicate a correlation between viral load and its consequences and diet, and the potential benefits of implementing clinical nutrition as a strategy to alleviate severe cases of CMS.

Clinical nutrition intervention was implemented in C3 after CMS diagnosis, when mortality had already surged significantly, reaching 1.5% of the total number of fish or approximately 300–400 fish per day. Such high mortality numbers make it untenable to continue production from both fish health and economic perspectives. But, already two weeks into the clinical nutrition period, we observed a gradual decrease in mortality, which further declined in the next six weeks. During this period there was a sudden spike in mortality coinciding with the relocation of fish in C3 to a different farm site due to logistical reasons. The handling and transportation of fish in well-boats are known to be major stressors, and at this stage, the fish in C3 still exhibited severe heart changes within the population. Hence, the increase in mortality following relocation was not surprising and underscores the importance of minimizing stress and avoiding handling of CMS diseased fish. However, approximately two weeks after relocation, the mortality rate again declined and became stable at an insignificant level until slaughter. The authors are not aware of any prior description of severe CMS outbreaks where the mortality rate has decreased to such a low level. We postulate that this outcome is attributable to the anti-inflammatory effects of the elevated levels of EPA and DHA in the feed, as indicated by the EPA and DHA incorporation into the heart tissue, as observed through fatty acid profiling. Previous studies have indicated the beneficial effects of high inclusion levels of EPA and DHA on viral diseases in Atlantic salmon such as heart and skeletal muscle inflammation (HSMI) and CMS [13,19]. In the latter study, they demonstrated a significant increase in EPA and DHA in heart phospholipids after feeding with a diet containing elevated levels of EPA and DHA. In the present study, we observed a corresponding increase in EPA and DHA in total lipids in the heart. Moreover, in another study it was shown that the heart and brain conserved EPA and DHA levels better than skeletal muscle, liver, skin, and intestine [20], highlighting the vital role of these fatty acids in the heart tissue of salmon. Also, the recent study by Mogahadam et al. [15], suggests that genetic control of PUFA production is important for PMCV resistance, which is in concordance with our results.

Histopathological scoring revealed substantial individual variation, but a representative sampling of ten fish has proven adequate in previous studies and in routine diagnostic work to provide comprehensive information on the health status of a cage population [12]. Furthermore, the consistency observed between the histopathological score, viral load, and mortality rate demonstrates the linking between these parameters and the reliability of each of them. High histopathological heart scores were documented in the population for the first two samplings, followed by a decreasing trend in the third sampling. Notably, the PMCV load was also extremely low at this stage, as confirmed by both RT-qPCR and ISH. The fish in C3, supported by clinical nutrition, presumably managed to clear the PMCV infection and regenerate heart damage. However, we did not see evident signs of regeneration in our study, nor did we investigate for it and thus this remains speculative. Future studies focusing on clinical nutrition and CMS should explore whether enhanced regenerative cardiac capacity plays a pivotal role in mitigating the disease and if supplemented omega-3 fatty acids can aid in this process.

PMCV RNA levels were investigated by both RT-qPCR and in situ hybridization. Overall, the results correlated well, showing the highest levels of PMCV RNA at sampling 2. Five samples from sampling 3 had Ct levels above the cut-off value and were thus regarded as PMCV negative by RT-qPCV. However, in situ hybridization revealed that virus RNA indeed was present within these samples, though in low amounts. This shows the high sensitivity of the method and the possible benefits of incorporating in situ hybridization as a diagnostic method for early and accurate detection of PMCV.

Controlled experimental challenge trials with PMCV do not reproduce the mortality observed in severe field outbreaks of CMS. Therefore, field studies on natural CMS outbreaks are complementary to controlled laboratory studies and necessary to obtain better knowledge of the pathogenesis of CMS, and thereby for the development of more effective measures against the disease. However, field studies have scientific challenges such as the inability to control environmental factors and often lack of adequate replicate sea cages. Within these limitations, the data presented in our study may nevertheless be an indicator for future investigations into the efficacy of diets enriched with EPA + DHA for mitigation of the severity of CMS outbreaks. In conclusion, this study highlights the potential benefits of clinical nutrition, specifically elevated EPA and DHA levels, in alleviating severe CMS outbreaks in farmed Atlantic salmon. It emphasizes the importance of stress reduction and careful handling during disease outbreaks and encourages further research into the role of enhanced cardiac regeneration in combating CMS.

## Figures and Tables

**Figure 1 microorganisms-12-00026-f001:**
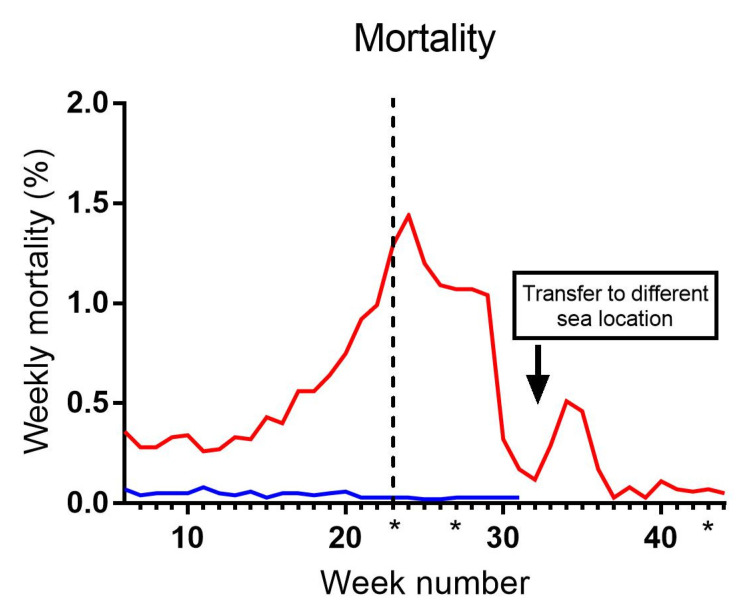
Weekly mortality (%) in sea cages 2 (blue) and 3 (red). The dotted line indicates a diet change to *Aqura* in C3 from week 23. Asterisks indicate the time of samplings (weeks 23, 27 and 43). Asterisks indicate the three sampling time points.

**Figure 2 microorganisms-12-00026-f002:**
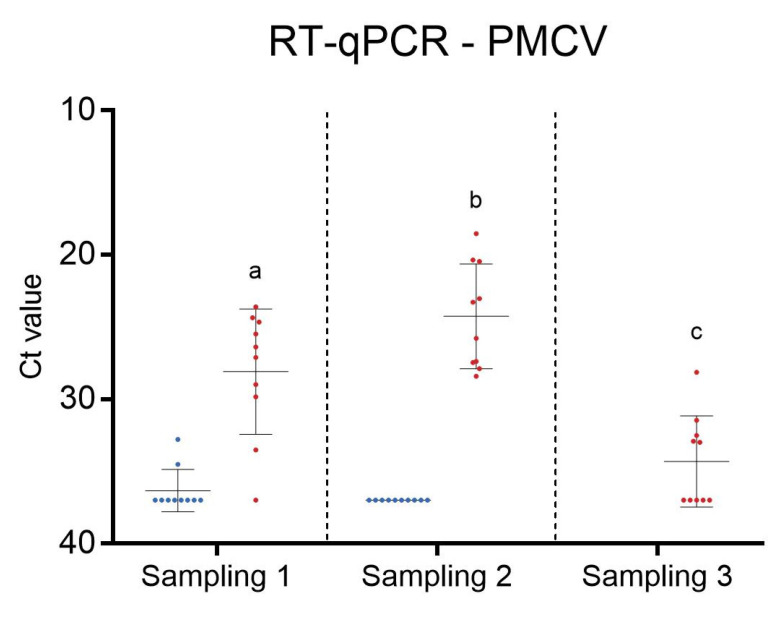
RT-qPCR for Piscine myocarditis virus (PMCV) in C2 (blue) and C3 (red) at different sampling time points. Dot plot with ten fish in each group at each sampling. The median value is indicated by a line. Levels not connected by the same letter (a, b, and c) are significantly different.

**Figure 3 microorganisms-12-00026-f003:**
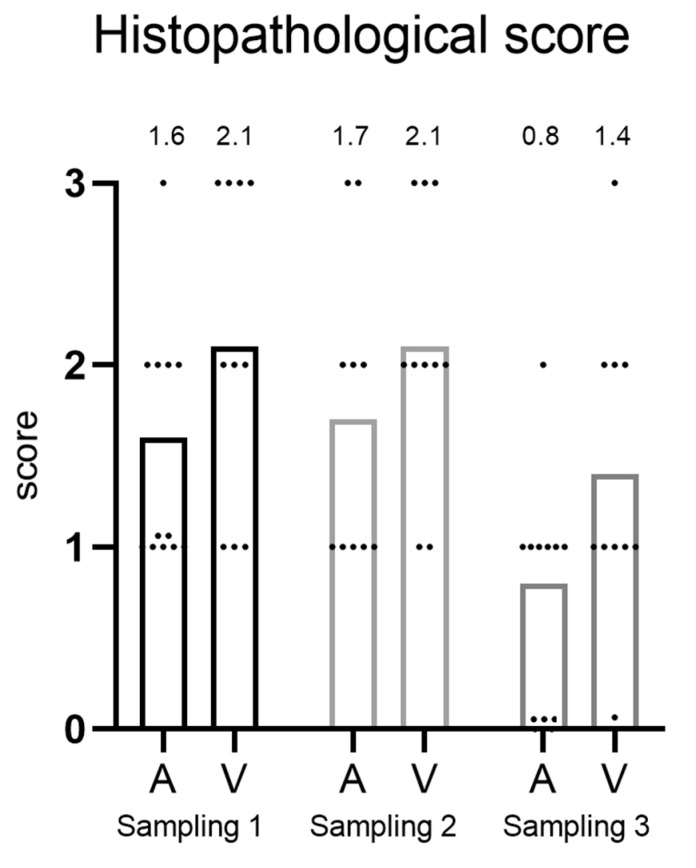
Histopathological score (from 0 to 3) of heart from fish in cage 3 at different sampling time points. A = atrium and V = ventricle. The mean score is shown above each bar.

**Figure 4 microorganisms-12-00026-f004:**
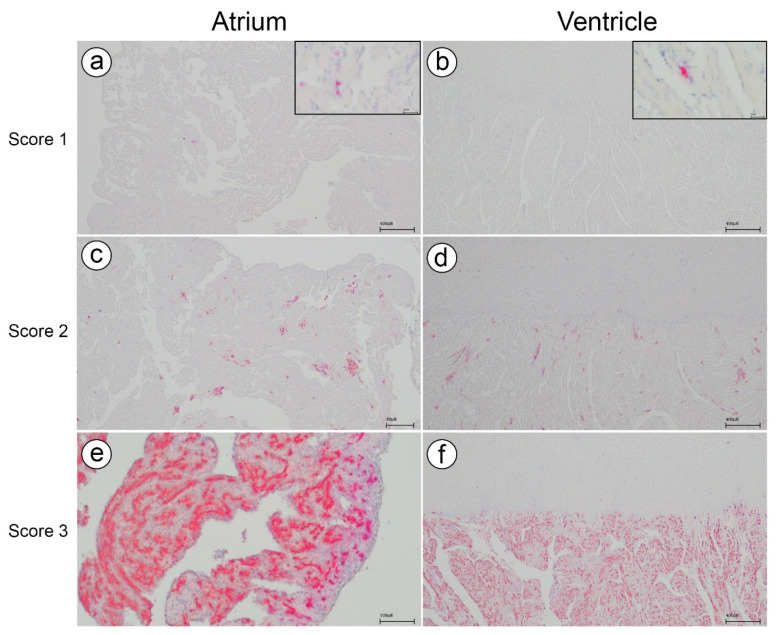
In situ hybridization for Piscine myocarditis virus (PMCV). Distribution of viral RNA in the atrium and ventricle across the different histopathological scores (1–3) in C3. (**a**,**b**) showing few and scattered positive cells (red) in the atrium and the spongious compartment of the ventricle, respectively. (**c**,**d**) showing multifocal positive cell clusters (red) in the atrium and the spongious compartment of the ventricle, respectively. (**e**,**f**) showing widespread and confluent positive cells in the atrium and the spongious compartment of the ventricle, respectively.

**Figure 5 microorganisms-12-00026-f005:**
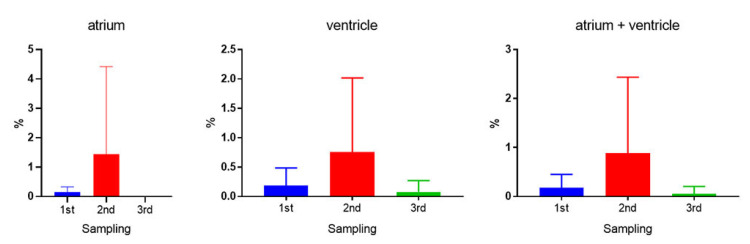
Quantification of PMCV RNA in the heart of fish in Cage 3. Area fraction virus (%) based on in situ hybridization targeting PMCV in the atrium, the ventricle and both compartments combined at the different sampling time points (first sampling—blue, second sampling—red, third sampling—green). The highest load was detected at the second sampling in both the atrium and the ventricle. The atrium was almost devoid of PMCV RNA (0.0019%) at sampling 3. The ventricle also had a low percentage of PMCV RNA at sampling 3 (0.08%).

**Figure 6 microorganisms-12-00026-f006:**
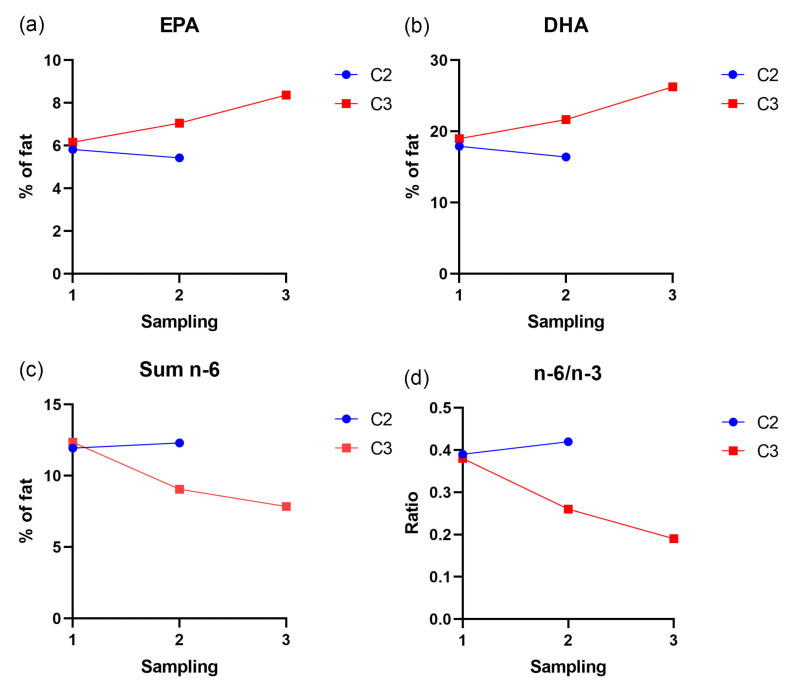
Fatty acid profile in heart tissue in pooled samples from C2 and C3. Each point represents values for a pooled sample of ten fish. (**a**) Shows the increase in EPA from sampling 1 to 3 in cage 3 (**b**) Shows the increase in DHA from sampling 1 to 3 in cage 3. (**c**) Shows the decrease in sum omega-6 in cage 3 from sampling 1 to 3. (**d**) Shows the decrease in n-6/n-3 ratio in cage 3 from sampling 1 to 3.

## Data Availability

Excess data beyond the results presented in the paper is available upon request to the corresponding author.

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
