# Peer review of "An Atypical Course of Cardiomyopathy Syndrome (CMS) in Farmed Atlantic Salmon (Salmo salar) Fed a Clinical Nutrition Diet"

_microorganisms, 2023, doi:10.3390/microorganisms12010026_

Round 1

Reviewer 1 Report

Comments and Suggestions for Authors

The present manuscript describes a well succeeded attempt to treat an atypical course of cardiomyopathy syndrome (CMS) in farmed Atlantic salmon (Salmo salar) with a clinical commercial diet, enriched with polyunsaturated fatty acids. Although the premise is considerably important for the salmon farming, the manuscript present serious flows regarding basic scientific methods. Although each of sampling had 10 fish per group, there was only two cages, one “healthy” and another presenting mortality, which makes the fish pseudo replicates and not replicates of each group. Moreover, 10 fish per cage por sampling is a number to low to be considered statistically representative. The problems of performing studies like this in fish farms, which are commonly called as “real world”, are the great risk of erroneous conclusions due to the lack of control. In this case, although the beautiful data showed by the authors, the use of only two cages (one for each group) improves too much the occurrence of type I error, the false positive, which is usually the most dangerous for science.

Nevertheless, it is impossible to simply discard the data presented by the authors, which are visibly fruit from hard work and may be used for further research. Thus, I recommend the resubmission of the manuscript as case study instead of scientific article if the journal have this modality. If not, I advise the authors to find a journal that publish case studies.

Best regards.

Comments on the Quality of English Language

There are little mistakes that can be easly improved.

Reviewer 2 Report

Comments and Suggestions for Authors

1. L96-118: The experimental design is unclear. How many cages participated in this study and how many cages were sampled? How many fish were collected in each cage? These should be written clearly.

2. L111-112: Why are the samples of histological analysis kept in 10% phosphate-buffered? This is wrong.

3. L185-192: Put it in L95 as section 2.1 is more appropriate.

4. L185-192: It is suggested to give the formula table of two kinds of feed to make readers know the difference between them.

5. L215: The significance labeling is a bit confusing in Figure 2.

Comments on the Quality of English Language

The language can be improved.

Round 2

Reviewer 1 Report

Comments and Suggestions for Authors

Dear Authors,

I am writing to you once again to extend my congratulations to your research team for the effort that has resulted in the present manuscript. I truly believe your work has great merit and should be published. Initially, I had my doubts, but in your response to the referees, it became clear that the authors are fully aware of the limitations associated with the performance of field research. Therefore, I think the manuscript can be accepted for publishing, provided the authors make the following crucial changes:

Sentences portraying the idea presented by "However, field studies have scientific challenges such as the inability to control environmental factors and often lack adequate replicate sea cages" should be incorporated into the abstract and the introduction section, not only in the discussion.

There is no doubt that the study is important and provides valuable information for the research field, particularly by showcasing data produced on a farm. However, it is of great importance to compel the reader to understand the limitations of any study conducted outside a controlled environment, and this one is no different.

Comments on the Quality of English Language

No comments.

Author Response

Dear Reviewer #1

Thank you for nice comments. We completely agree and have changed the abstact and the introduction accordingly:

Abstract:

Cardiomyopathy syndrome (CMS) poses a significant threat to farmed Atlantic salmon (Salmo salar), leading to high mortality rates during the seawater phase. Given that controlled experimental challenge trials with PMCV do not reproduce the mortality observed in severe field outbreaks of CMS, field trials on natural CMS outbreaks are warranted. This field study explored the impact of a clinical nutrition intervention, specifically a diet enriched with eicosapentaenoic acid (EPA) and docosahexaenoic acid (DHA), on a severe CMS outbreak in a commercial sea farm. CMS was diagnosed in a single sea cage with high mortality rates. Histopathological analysis, RT-qPCR and in situ hybridization for virus detection, and fatty acid composition analysis were used to monitor the impact of disease and the inclusion of EPA and DHA in heart tissue. Following implementation of clinical nutrition, a decline in mortality rates, regression of CMS-associated changes, and a significant reduction in piscine myocarditis virus (PMCV) RNA load were observed within the salmon population. Fatty acid composition analysis of heart samples demonstrated increased levels of EPA and DHA, reinforcing the association between dietary factors, viral load dynamics, and overall fish health. Although further validation is needed in future studies, as field trials may not be sufficient to establish causation, our results indicate that optimizing EPA + DHA levels may prove beneficial in severe CMS outbreaks.

Introduction, last paragraph:

In this descriptive field study, we report findings from the use of clinical nutrition in large Atlantic salmon (approximately 2.5 kg and fed until slaughter weight) affected by a CMS outbreak at a typical commercial sea farm. Using RT-qPCR and in situ hybridization for PMCV detection, histopathological analysis, and lipid composition analysis of the heart, we demonstrate decline in mortality, overall regression of CMS-related changes, and reduction of viral RNA load within the population. We found a significantly altered fatty acid composition in the heart in fish fed the clinical nutrition diet. The findings are significant as they contrast the typical progression of severe CMS outbreaks, where premature slaughter is the only economical viable option. Although further validation is needed in future studies, as field trials may not be sufficient to establish causation, our results emphasize the importance of optimizing EPA + DHA levels in order to mitigate CMS and thus generally improve fish health in the later stages of the marine growth phase.